# Feasibility Study of Dose Modulation for Reducing Radiation Dose with Arms-Down Patient Position in Abdominal Computed Tomography

**DOI:** 10.3390/diagnostics12020323

**Published:** 2022-01-27

**Authors:** Jina Shim, Yong Eun Chung, Hyun-Woo Jeong, Youngjin Lee

**Affiliations:** 1Department of Diagnostic Radiology, Severance Hospital, 50-1, Yonsei-ro, Seodaemun-gu, Seoul 03722, Korea; eoeornfl@korea.ac.kr; 2Department of Radiology, Yonsei University Health System, Yonsei University College of Medicine, 50-1, Yonsei-ro, Seodaemun-gu, Seoul 03722, Korea; yelv@yuhs.ac; 3Department of Biomedical Engineering, Eulji University, 553, Sanseong-daero, Sujeong-gu, Seongnam-si 13135, Gyeonggi-do, Korea; 4Department of Radiological Science, Gachon University, 191, Hambakmoero, Yeonsu-gu, Incheon 21936, Korea

**Keywords:** abdominal computed tomography, arms-down position, radiation dose reduction, dose modulation, evaluation of image quality and dose

## Abstract

This study was carried out to demonstrate whether the radiation dose for patients in arms-down position can be reduced without affecting the diagnosis on abdominal computed tomography (CT). The patients were divided into two groups: group A, which included patients with arms-down position using dose modulation on, and group B, which included patients with arms-down position using dose modulation turned off. Quantitative evaluation was compared using Hounsfield units, standard deviation, and signal-to-noise ratio of the four regions. The qualitative evaluation was assessed for overall image quality, subjective image noise, and beam hardening artifacts. Dose evaluation for CT dose index (CTDI) and dose length product (DLP) was compared by comparing the CT images with dose modulation turned on and off. In the quantitative and qualitative evaluation, there was no statistically significant difference between groups A and B (*p* > 0.05). In the dose evaluation, the CT images with dose modulation turned off had significantly lower CTDI and DLP than the CT images with dose modulation turned on (*p* < 0.05). Our results suggest that, for the GE Revolution EVO CT scanner, turning off dose modulation and increasing the tube voltage can reduce the radiation dose for patients with the arms-down position without affecting the diagnosis. This study did not consider the change of tube potential according to the use of dose modulation, and we plan to conduct additional research in the future.

## 1. Introduction

Computed tomography (CT), which was first introduced in 1972, has been developed for a variety of clinical applications and is now considered an essential diagnostic imaging equipment. Further, multidetector CT (MDCT) has become an indispensable imaging device in the clinical setting. With the development of MDCT, it is now possible to acquire images of many more slices simultaneously, expanding the clinical usefulness even more than before.

The development of MDCT allowed for the scanning of more regions in less time and increased the resolution of the longitudinal axis. In particular, the 64-slice MDCT can acquire thinner cross-sections, thereby obtaining isotropic volume data. Isotropic volume data has the advantage of allowing the user to reconstruct an arbitrary cross-sectional image and view a 3D image. However, to obtain isotropic volume data, a thin cross-sectional image must be acquired, which increases image noise. There is an inevitable disadvantage of increasing the radiation dose in order to improve the image quality, which has decreased due to the increased noise. MDCT is widely used in the clinical setting, but it increases the radiation dose to the patient [1,2]. This increased radiation dose from MDCT raises concerns about increased cancer risk and is a current public health issue [2,3].

One of the methods of reducing the radiation dose is dose modulation [4,5,6]. This technology automatically adjusts the tube current in two directions (x–y-axis and z-axis) by calculating the difference in radiation attenuation in the body while maintaining an acceptable image quality [7,8,9]. Using the dose modulation technique, it is possible to obtain a constant image quality regardless of patient size. Previous studies reported that the use of dose modulation can reduce the radiation dose up to 60% compared to the fixed tube current [10,11]. In underweight or overweight patients, this technique can obtain diagnosable images while minimizing the radiation dose. In addition, dose modulation has been shown to reduce the radiation dose in both pediatric and adult patients [12].

Although it has been reported that the use of dose modulation has radiation dose benefits for patients, improper use may result in disadvantages [7,9]. A previous study found that the inclusion of dense substances within the scan range exposes the patients to more radiation than is necessary [13]. This shows that dose modulation was improperly used owing to surrounding factors rather than patient factors. CT scans are not always performed in a consistent and appropriate environment due to the high volume of emergency patients in the emergency room. In particular, trauma patients are sometimes examined in the arms-down position, resulting in low image quality and high radiation doses [14,15]. However, it is debatable whether lowering the patient’s radiation dose by manually raising the hand of an emergency patient should be a higher priority in an emergency situation. The better image with a lower dose could be produced by raising the arms, but if the hand of an unconscious patient is not fixed, it can lead to a safety accident due to a moving table during the CT scan. Therefore, this study determined whether abdominal CT scans without dose modulation for the GE Revolution EVO CT scanner are beneficial in terms of image quality and radiation dose to patients in an arms-down position.

## 2. Materials and Methods

### 2.1. Patients

This retrospective study was approved by the institutional review board of Severance Hospital, and the requirement for informed consent was waived due to the retrospective design of the study. In our hospital, patients who can communicate during abdominal CT are instructed to raise their arms for examination before the CT scan. However, in the emergency room, patients with drowsy mental status and difficulties in communication were examined to keep their arms raised. This retrospective study evaluated adult patients who underwent CT examination in the emergency room (approximately three months apart), including noncontrast and hepatic venous phase (HVP) CT, between 2018 and 2019. Group A was examined using the existing protocol; a total of eight patients (male:female ratio, 3:5) with a mean age of 68.3 ± 19.3 years (range, 42–75 years) and mean body mass index (BMI) of 24.4 ± 6.1 kg/m^2^ (range, 16.18–31.87 kg/m^2^) were selected. The number of underweight patients were 2, healthy weight 2, overweight 2, and obesity 2. Group B was examined with the dose modulation turned off; a total of 23 patients (male:female ratio, 12:11) with a mean age of 73.3 ± 14.1 years (range, 37–95 years) and mean BMI of 24.1 ± 5.8 kg/m^2^ (range, 15.6–37.4 kg/m^2^) were retrospectively analyzed for imaging and radiation dose evaluation. The number of underweight patients were 3, healthy weight 14, overweight 2, and obesity 4.

### 2.2. CT Acquisition and Radiation Dose Measurements

All abdominal CT examinations were performed using a 64-detector CT scanner (Revolution EVO; GE Medical Systems, Milwaukee, WI, USA). All CT scans ranged from 1 cm superior to the diaphragm to the lesser trochanters. HVP CT was obtained 55 s after the attenuation in the abdominal aorta increased to 100 Hounsfield unit (HU) compared to the baseline. When HVP phase images were obtained, an intravenous injection of 2.0 mL/kg (up to a maximum 150 mL when patients weighed more than 75 kg) of iodinated contrast media (Omnipaque 300 (Iohexol), GE Healthcare, Cork, Ireland), followed by a bolus injection of 40 mL of saline chaser, was performed. The scan parameters used in groups A and B are listed in Table 1. In both groups, abdominal noncontrast CT was performed at 100 and 120 kV in order to reduce beam hardening artifacts. The mA range with dose modulation turned on and off was 100–350 in group A and fixed 170 in group B. The reason for setting the fixed 170 mA is that the radiation dose is increased by 2.5 times due to the tube voltage increased from 100 to 120 in the group B [16]. In order to reduce this, 50% of the maximum mA set in the smart mA was used. The scan parameters for HVP CT were identical for both the groups. The reconstruction algorithm used is filtered back projection algorithm. The iterative reconstruction (IR) was not used to exclude the effect on the reduction of image artifact.

In group B, the CT dose index (CTDI) volume and dose length product (DLP) were recorded in the noncontrast CT with dose modulation turned off and HVP CT with dose modulation turned on. To compare the radiation dose based on dose modulation, CTDI and DLP on noncontrast CT with dose modulation turned off and HVP CT with dose modulation turned on were compared in the same patient in group B. Since dose comparison is greatly influenced by the conditions, such as patient position and BMI, the same patient was evaluated with images of non-contrast and HVP without position change in order to minimize these variables.

### 2.3. Quantitative and Qualitative Image Analysis

All quantitative evaluations were performed using ImageJ 1.37v for Windows. Figure 1 is a schematic diagram showing the comparison targets for quantitative and qualitative evaluation. For quantitative evaluation, the noncontrast images of groups A and B were compared. Noncontrast CT is an examination that can be performed even by patients with side effects of contrast agents, and there are various indications that can be diagnosed. In this study, the image evaluation and comparison were performed with a noncontrast image only, but the image change that could occur by lowering the hand will be applied similarly to the contrast CT image, so it will be possible to interpret the phase contrast CT. One radiologic technologist (with 10 years of experience in radiology) measured the Hounsfield unit and objective image noise, which was defined as the standard deviation (SD) of the mean CT number in the liver, abdominal aorta, paraspinal musculature, and subcutaneous fat. The regions of interest (ROIs) were located in homogeneous regions, and the liver was located away from the vessel. The average size of ROI in the liver, abdominal aorta, paraspinal musculature, and subcutaneous fat were 21.51, 46.31, 224.79 and 44.24 mm^2^. The shape and size of the ROI for each anatomical location were made identical using the copy-and-paste function of the ImageJ software. The signal-to-noise ratio (SNR) was calculated according to the formula:$$\mathrm{SNR}=\frac{{\mathrm{HU}}_{\mathrm{value}}}{{\mathrm{SD}}_{\mathrm{value}}}$$

A qualitative evaluation of the CT images was performed by a board-certified radiologist with 10 years of experience in abdominal CT. As shown in Figure 1, for qualitative evaluation, the scores obtained from image evaluation were compared between groups A and B. Further, the noncontrast CT images of each group were scored by comparing them with the HVP CT images of the same patient, and the scores of groups A and B were compared. In the qualitative evaluation, images were reviewed for three weeks, and the image parameters were blinded through the picture archiving and communication system. All images were displayed side by side for comparative analysis of the same window width and level (350/50). The scores for beam hardening artifacts, overall image quality, and image noise were graded as follows: a score of 1 indicates the presence of much more beam hardening artifacts (overall image quality or image noise) than HVP CT, 2 indicates slightly more beam hardening artifacts than HVP CT, 3 indicates beam hardening artifacts similar to HVP CT, 4 indicates beam hardening artifacts slightly lesser than HVP CT, and 5 indicates beam hardening artifacts much lesser than HVP CT. Beam hardening artifacts on noncontrast CT were graded as follows: 1, much more beam hardening artifacts; 2, slightly more beam hardening artifacts; 3, similar beam hardening artifacts; 4, slightly less beam hardening artifacts; and 5, much less beam hardening artifacts.

### 2.4. Statistical Analysis

All data were analyzed using SPSS Statistics 19.0 (SPSS Inc., Chicago, IL, USA). The quantitative evaluation measures, HU, SD, and SNR, were analyzed using the Wilcoxon signed-rank test to determine the difference between the two groups. Qualitative evaluation was also performed using the Wilcoxon signed-rank test. *p* < 0.05 was considered statistically significant. To understand the significance of the scores in qualitative assessment, the frequency of scores obtained from the beam hardening artifacts, overall image quality, and the image noise were analyzed in terms of percentage.

## 3. Results

### 3.1. Image Quality

The reasons for CT examinations in these patients were as follows: follow-up of underlying malignancy (n = 8; 2 cases of pancreas head cancer, 1 cases of lung cancer, 2 cases of hepatocellular carcinoma (B-viral), 1 cases of duodenal carcinoid tumor, and 2 cases of prostate cancer), shock (non traumatic) (n = 3), fever (n = 7), edema (n = 2), stone (n = 2), dyspnea (n = 2), mental change (n = 1), traffic accident (n = 2), and abdominal pain (n = 4). In these patients, image quality and radiation dose were evaluated without raising their arms.

Quantitative evaluation was measured as HU and objective image noise, which was defined as the SD of the mean CT number and SNR in the liver, abdominal aorta, paraspinal musculature, and subcutaneous fat. As shown in Table 2, there was a statistically significant difference between groups A and B in the SD of the liver (lower in group B) and HU of subcutaneous fat (higher in group B) (*p* < 0.05). The abdominal aorta and paraspinal musculature did not show statistically significant differences for HU, SD, and SNR. In addition, liver showed no statistically significant difference for HU and SNR, and subcutaneous fat for SD and SNR (*p* > 0.05).

For qualitative evaluation, image beam hardening artifacts, overall image quality, and image noise on HVP and noncontrast CT were compared, and beam hardening artifacts were evaluated only on noncontrast CT (Figure 2). There was no statistically significant difference between groups A and B in the qualitative evaluations (*p* = 0.914, 1.000, 0.257, and 0.705) (Table 3). The qualitative evaluation of scores obtained by comparing beam hardening artifacts, overall image quality, and image noise with HVP CT was reported in terms of percentage. Since the number of samples in groups A and B was different, it was expressed as a percentage, not the number of patients corresponding to each score. In the beam hardening evaluation, group A had the highest proportion of two or fewer points (87.50%), and group B had the highest proportion of three or higher points (21.74%). In the overall image quality evaluation, group A had the highest proportion of two or fewer points (87.50%), and group B had the highest proportion of three or higher points (21.74%). In the image noise evaluation, group A had the highest proportion of two or fewer points (87.50%), and group B had the highest proportion of three or higher points (17.39%). The qualitative evaluation of scores obtained from the evaluation of beam hardening on noncontrast CT was reported in terms of percentage. In this evaluation, group A had the highest proportion of two or fewer points (37.50%), and group B had the highest proportion of three or higher points (73.91%).

### 3.2. Radiation Dose Measurements

CTDI and DLP on noncontrast and HVP CT images were compared in group B. As shown in Figure 3, a statistically significant difference was observed when the radiation dose on noncontrast CT without dose modulation was compared with that of HVP CT with dose modulation turned (*p* < 0.05). The CTDI ranged from 6.43 to 8.18 mGy (mean: 7.48 mGy) in noncontrast CT and 4.46 to 15.48 mGy (mean: 11.78 mGy) in HVP CT. The DLP ranged from 344.5 to 483.64 mGy (mean: 415.31 mGy) in noncontrast CT and 208.53 to 930.46 mGy (mean: 661.15 mGy) in HVP CT. Compared to the radiation dose on HVP CT with dose modulation turned on, the average dose reduction in noncontrast CT with dose modulation turned off was 57% for CTDI and 59% for DLP.

## 4. Discussion

The purpose of this study was to confirm the usefulness of turning off dose modulation in lowering the radiation dose while maintaining diagnostic image quality in patients who are unable to raise their arms. Previous studies have reported that raising the arm can reduce the effective attenuation in patients, thereby reducing artifacts and lowering the radiation dose [15,17]. Although raising the arm has the advantage of making an improved image with reduced artifacts, raising the arm artificially of an unconscious patient can take a lot of time and lead to a safety accident. Another study reported that a patient with arm repositioning experienced anterior glenohumeral dislocation and consecutive plexus injury [18]. Hence, studies have been conducted to obtain diagnosable images without artificially repositioning the patient’s arm. Using IR reduced the beam hardening artifacts caused by positioning the arms next to the torso, as well as the radiation dose [19]. However, IR could create an artificial texture and poor image sharpness, resulting in deteriorated image quality [20]. In addition, it cannot be applied to all cases because not all CT equipment can use advanced techniques such as IR. In this study, the usefulness of dose modulation turned off was confirmed to perform an appropriate abdominal CT scan without raising the arms of patients in all situations.

In a study of 177 patients conducted in 2008 [14], the CT images of patients with the arms-down position had lower image quality despite the high radiation dose. In this study, there was a difference in noise in the liver region between patients who had their arms down and those who had their arms raised. Kahn et al. [21] reported that artifacts caused by the arm interfered with the interpretation of liver hemorrhage. Another study reported that images of upper abdominal organ regions in patients with the arms-down position showed severe artifacts [15]. Since the liver region overlaps with the distal humerus, it has the greatest total effective thickness when the hand is lowered [22]. Therefore, the image noise is increased because of the reduced number of photons in the liver (Figure 4). The results of the quantitative evaluation in our study showed that the image noise in the liver region decreased in group B. Image noise can be lowered by increasing the tube voltage for photons that have been reduced due to the increased effective thickness. Further, there was no statistically significant difference in the mean values of groups A and B; however, differences in percentage according to the evaluation criteria were observed. In the four qualitative image evaluation criteria, group B received fewer scores of 2 or less and more scores of 3 or more than group A. The image in group B was not qualitatively inferior to that in group A. It was confirmed that the diagnostic acceptability for group A could be also maintained in group B, although it could not be concluded that the quality of the entire abdominal image in group B was improved than that of group A only by reducing liver region noise.

A previous study reported that raising the arm was more beneficial in terms of radiation dose than lowering it [23]. However, this study evaluated the usefulness of turning off dose modulation in reducing the radiation dose in trauma patients who are unable to raise their arms while maintaining image quality. A comparison of the radiation dose showed that the dose in the patient with dose modulation turned on was increased. Because both arms were included in the scan range, the effective attenuation increased, and the increased effective attenuation resulted in a high dose when dose modulation was used. The radiation dose range is wider when the dose modulation is turned on than when it is turned off. When dose modulation was used, the radiation dose was adjusted to maintain image quality according to BMI. There is a potential effect of obtaining different results from this study, when the fixed tube current is used like group B in patients with high BMI. However, there was no significant difference in the radiation dose proportional to BMI because, when calculating the effective thickness using a scout image, the calculation weight varies depending on the order or frequency of position, such as anteroposterior and lateral scout views [24]. Therefore, the effective attenuation calculated from the scout image may vary depending on the position of the arms (for example, on the sides and on the stomach), which may have affected the irradiation.

The difference in the average dose reduction was larger in DLP than in CTDI. DLP is calculated by multiplying the scan length in the CTDI, and the scan length involves the subjective judgment of the radiologic technologist based on the patient’s condition during the examination. In general, patients with their arms down during CT are often those who are unable to communicate, and it is difficult to control their breathing. Since abdominal CT should be performed up to the liver dome, the scan length in these patients should be set more conservatively than in other patients. Such a conservative scan length may increase the dose difference between DLP and CTDI.

In this study, the dose and image quality were evaluated when dose modulation was not applied during abdominal CT examination of patients with difficulty raising patient’s arms. When the arm is down during abdominal CT examination, the image quality deteriorates due to beam hardening artifacts, and the effect of changes in tube voltage and tube current conditions on the image is also very important. Jeon et al. confirmed that the quantitative evaluation of noise level was worse than the general situation during abdominal emergency CT scan with both arms lowered or only one arm raised after a motor vehicle accident [25]. In the above research team, when the tube voltage, which has a great influence on beam hardening, was increased from 80 to 140 kVp, the image noise was reduced. In particular, it was proved that noise was greatly increased in all tube voltage conditions when dose modulation was applied to CT scan after lowering the arm. By applying the reference results in addition to the results of this study, we expected that it would be possible to suggest a new approach in various conditions of abdominal scan.

In CT examination, dose modulation can lower the exposure by controlling the optimal tube current that can form an image quality useful for diagnosis. In the study of Kalra et al., when dose modulation was used, a dose reduction effect of 10 to 43% was obtained at a noise index of 12 to 15 [26]. However, in this study, when the dose modulation was not used in the arm-down position of the patient during CT examination, we found that dose reduction and image quality improvement could be achieved at the same time. These results did not consider the tube voltage change according to the use of dose modulation, and we intend to analyze the results by conducting a study of the same design considering the tube voltage change in the future.

This study has some limitations. Firstly, the number of subjects was small and asymmetric due to retrospective evaluation. Secondly, since repeated acquisition of CT scans in the same patient is unethical, the patients were divided into two groups, and different patients were scanned at differences phases. However, in order not to obtain biased results as intended by the authors, the two groups of patients were randomly selected. In addition, the only difference between the images of the non-contrast and HVP phases of the same patient was the presence or absence of a contrast agent in the body. The difference in radiation dose based on the attenuation between the contrasted image and the non-contrast image was ignored because it was expected to be insignificant. Thirdly, the image quality was quantitatively evaluated based on a single point metric. Finally, the results of this study cannot be generalized, because it evaluated only one vendor and was reviewed by one radiologist. This paper showed the possibility of reducing the dose while maintaining the existing image quality in patients with arm down in a simple and easy way through the GE Revolution EVO CT scanner.

## 5. Conclusions

The purpose of this study was to confirm the feasibility of turning off dose modulation to lower the radiation dose while maintaining diagnostic image quality in patients who are unable to raise their arms on abdominal CT. Our results suggest that, for the GE Revolution EVO CT scanner, turning off dose modulation and increasing the tube voltage can reduce the radiation dose without affecting the diagnosis for patients undergoing abdominal CT in the arms-down position.

## Figures and Tables

**Figure 1 diagnostics-12-00323-f001:**
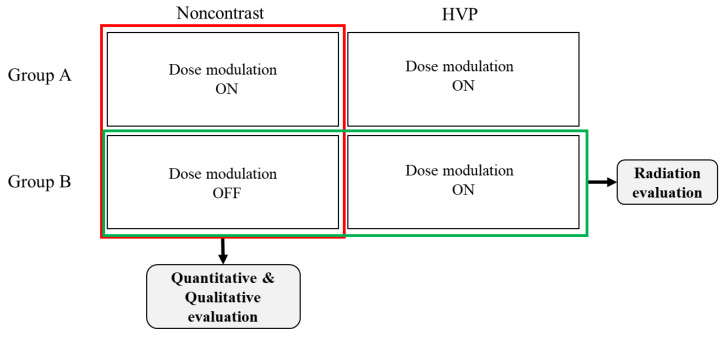
Schematic diagram of study design showing the comparison target for quantitative, qualitative, and radiation dose evaluation.

**Figure 2 diagnostics-12-00323-f002:**
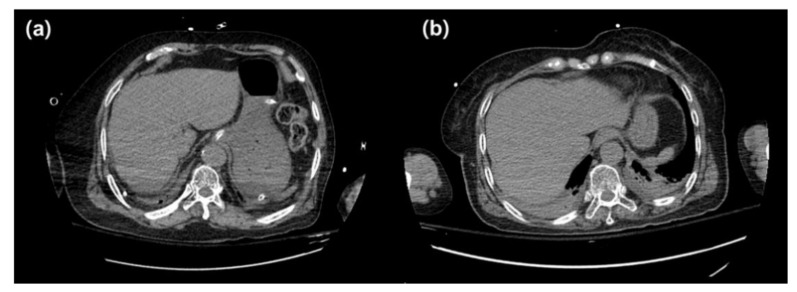
Abdominal CT images of patients with arms-down position in groups A and B. (**a**) Axial image in a group A patient (M, 75 years, BMI: 26.12 kg/m^2^), (**b**) axial image in a group B patient (F, 69 years, BMI: 24.44 kg/m^2^). Although beam hardening artifacts appear in groups A and B due to the arm, it would be seen that less beam hardening artifacts appear in the dose modulation off image in patients within the normal BMI range.

**Figure 3 diagnostics-12-00323-f003:**
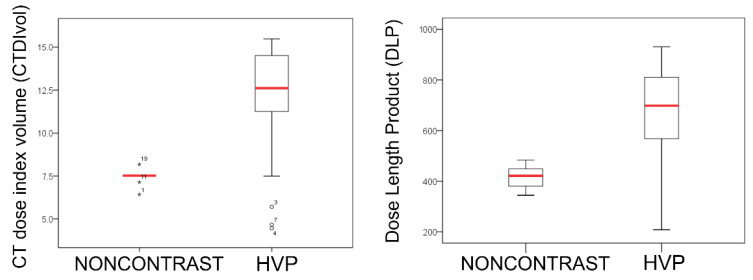
Graphs of CTDI volume and DLP in group B, with dose modulation turned on (HVP) and off (noncontrast).

**Figure 4 diagnostics-12-00323-f004:**
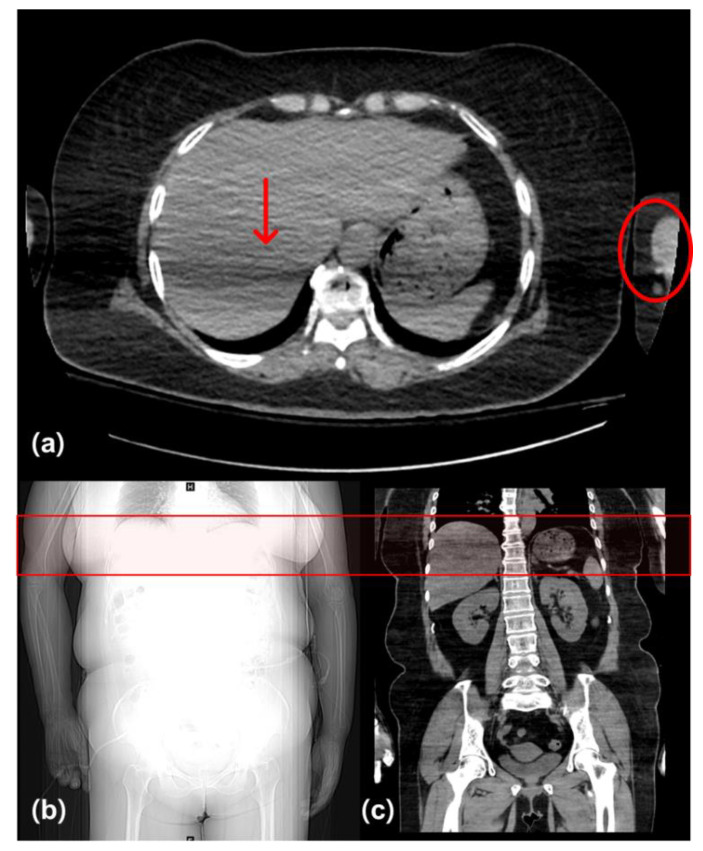
Abdominal CT images with an artifact (red arrow) due to overlap of the liver region with the distal humerus (red circle). This region has the greatest total effective thickness because of the arm (red box). (**a**) Axial image, (**b**) scout image, and (**c**) coronal image of a group A patient (M, 54 years, BMI: 31.87 kg/m^2^) with arms-down position. Values of CTDI and DLP were 9.60 mGy and 545.37 mGy·cm, respectively.

**Table 1 diagnostics-12-00323-t001:** Abdominal noncontrast and HVP CT scan parameters for groups A and B.

	Group A Noncontrast	Group B Noncontrast	Group A and B HVP
**Tube potential (kV)**	100	120	120
**Beam collimation (mm)**	2.5	2.5	0.625
**mAs (mAs)**	Smart mA (100~350)	170	Smart mA (100~350)
**Beam pitch**	0.984	0.984	0.984
**Rotation time (s)**	0.5	0.5	0.5
**Scanning direction**	Feet first	Feet first	Feet first
**Delay (s)**	-	-	55
**Reconstructed section thickness (mm)**	3	3	3
**Section overlap (mm)**	3	3	3
**Noise index**	17.86	-	22.52

**Table 2 diagnostics-12-00323-t002:** Image noise and HU in liver and subcutaneous fat of Group A (dose modulation on) and Group B (dose modulation off).

	Group A	Group B	*p* Value
**SD of liver**	23.73 ± 4.11	21.51 ± 5.01	0.036
**HU of subcutaneous fat**	−120.83 ± 8.79	−99.41 ± 34.06	0.017

**Table 3 diagnostics-12-00323-t003:** Subjective image quality scores of Group A (dose modulation on) and Group B (dose modulation off). Data are mean ± standard deviation.

	Beam Hardening Artifacts	Beam Hardening Compared to HVP	Overall Image Quality	Image Noise
**Group A**	2.75 ± 0.66	2.12 ± 0.33	1.75 ± 0.66	1.87 ± 0.59
**Group B**	2.86 ± 0.84	2.04 ± 0.62	2.04 ± 0.62	1.95 ± 0.62

## Data Availability

Not applicable.

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
