# Peer review of "Feasibility Study of Dose Modulation for Reducing Radiation Dose with Arms-Down Patient Position in Abdominal Computed Tomography"

_diagnostics, 2022, doi:10.3390/diagnostics12020323_

Round 1
Reviewer 1 Report
Compliments for upgrade of your manuscript.
The added notes allow a better understanding of the research methodology employed.
Author Response
Thank you for review and comment in this manuscript.
We have revised the paper as your suggestion and responded point by point.
Please confirm attached revised manuscript and response files.
Best regards,
Youngjin Lee

Reviewer 2 Report
Overall an interesting study which investigated the effects of using tube current modulation when patients arms are left down during abdominal CT. As expected using lower set mAs resulted in significant dose reductions but using different kV values between protocols made it difficult to compare image quality wise. It would additionally be useful to report the actual mAs ranges used when modulation was turned on for patients so the effects of dose modulation can be seen clearly. Discussion should also evaluate the fact that dose modulation may increase the tube current in the upper abdominal areas where the elbows and forearms lie but potentially reduce the tube current too much in the pelvic area for diagnostic purposes but this has not been evaluated or commented on
Not clear here whether iterative reconstruction was used during the CT protocol which again could have reduced artefacts
Hard to compare protocols directly given that different tube voltages have been used for each group
Abstract
Background
Materials and methods
Were the same patients scanned twice here?
Why was mental change an acceptable reason for an abdominal CT?
Very unclear how patients were assigned to groups (? Random)
Patient characteristics is better reported in the results section
Group B has some larger patients included – did this influence the mean doses reported (ie BMI up to 37.4)
Was radiation dose simply read out from the CT display? Was this calibrated recently and shown to be within acceptable limits?
Why is the noise index for Group A non contrast set lower than for the HVP protocol?
Was non-contarst CT used in all patients for your Abdomen CT examination? This should be justified
Hounsfield unit – not House unit pg4 line 135
Clearly state how SNR was calculated – i.e provide the formula used
Results
In Table 2 why only present results for two areas (liver and subcutaneious fat) – as all should be presented
Presenting results of the qualitative evaluation of scores as percentages is simplistic and entire results should be presented in a table
Discussion: Mostly summarises other work rather than cirtically evaluating the results presented here ijn this study. In particular further commentary is required on the different protocols used esp the influence of higher kV in Group B which will offset beam hardening artefacts and image noise
Conclusion: Is not completely accurate as neglects the influence of kilovoltage on image quality also
Images: What is the purpose of Figure 4? Better to present examples of artefacts from each of the protocols
Author Response

(The authors gave the same response as above.)

Round 2
Reviewer 2 Report
Although the conclusion has been updated to include reference to the impact of kilovoltage this has not been included in the abstract - and should be.
It is important to report the patient size characteristics in the results and disuss the potential impact of these differences on your results in the discussion.
The authors conclude that image quality is improved using this technique - based on statistically significant changes in the liver SD between groups A and B - but no difference in SD, HU, SNR was detected for the abdominal aorta, paraspinal musculature or liver and there were no differences in qualitative evaluation either - making it very difficult to conclude overall image quality is improved. Firstly this single location change in SD needs to be discussed and why changes were not noted in the other locations. Secondly the authors must be clear in their assertions that image quality is improved based on a single point metric when the rest remained the same. Thirdly the clinical effect of this change deserves comment (namely will a 2HU change in SD be noticed clinically).
Author Response
Dear reviewer,
Thank you for your valuable review comments.
Please check attached response and revised manuscript files.
Kind regards,
Youngjin Lee

Round 3
Reviewer 2 Report
Changes have improved the manuscript although grammar and readability need to be checked prior to publication